# The Impact of Nitrogen on the Yield Formation of *Artemisia dubia* Wall: Efficiency and Assessment of Energy Parameters

**DOI:** 10.3390/plants12132441

**Published:** 2023-06-25

**Authors:** Gintaras Šiaudinis, Algirdas Jasinskas, Danutė Karčauskienė, Regina Skuodienė, Regina Repšienė

**Affiliations:** 1Vėžaičiai Branch of the Lithuanian Research Centre for Agriculture and Forestry, Gargždų 29, LT-96216 Vėžaičiai, Lithuania; danute.karcauskiene@lammc.lt (D.K.); regina.skuodiene@lammc.lt (R.S.); regina.repsiene@lammc.lt (R.R.); 2Department of Agricultural Engineering and Safety, Faculty of Engineering, Agriculture Academy, Vytautas Magnus University, Studentu Street 15A, LT-53362 Akademija, Lithuania; algirdas.jasinskas@vdu.lt

**Keywords:** *Artemisia dubia*, nitrogen, DM yield, yield parameters, nitrogen utilization efficiency, energetical analysis, chemical and calorific values

## Abstract

With the increasing importance of energy crops, research on potential energy crops is carried out to identify plant species with high productivity and energy value. The field experiment with the new promising energy crop, *Artemisia dubia* (wormwood), was executed at the Vėžaičiai Branch of the LAMMC. The soil site was naturally acidic *Retisol* (pH 4.2–4.4). The species was investigated as an energy crop through the evaluation of its biomass productivity and some energetical qualities. According to the three investigation years, DM yield significantly varied depending on the growing season, cutting time and nitrogen rate. The highest average DM yield was observed in 2020—10.58 t ha^−1^. On average, the DM yield varied from 6.49 t ha^−1^ (first cutting) to 11.82 t ha^−1^ (third cutting). The DM yield was positively correlated with stem height and the mass of one stem. Nitrogen use efficiency (NUE) depended on the growing season, cutting time and nitrogen rate. Both N90 and N180 rates should be used for *A. dubia* fertilization. Energy growing analysis (including direct and indirect expenses) revealed that the highest share of energy expenses are due to indirect energy expenses (particularly nitrogen application). EUE (energy utilization efficiency) tends to decrease as a result of increasing nitrogen fertilization. Overall, *A. dubia* granules are characterized by a high calorific value.

## 1. Introduction

To mitigate climate change and reduce the importance of fossil fuels, the role of plant biomass in overall energy production is increasing every year. In 2018, EU leaders have set a goal to achieve the share of energy from renewable sources making up 32% of the final energy consumption of the European Union by 2030 [1]. In the long term, the successful propagation of energy crops should ensure species diversity, save non-renewable resources, additional jobs and income, sustainability and energy security [2,3]. Energy crop studies are being conducted at research institutions in various countries to identify and evaluate plant species suitable for different types of biofuels, focusing on local and non-native species, and their ability to produce high annual biomass yields with their high energetic value [4,5].

At the Lithuanian Research Centre for Agriculture and Forestry (LAMMC), research on different local and introduced energy crop species has been conducted over a period of two decades. *Artemisia dubia* Wall. (wormwood) is among the plant species studied. Plants of the *Artemisia* genus are classified within published US and British sources as plants of the future due to their wide application in the food industry and medicine [6,7]. *A. dubia* L. (wormwood) as a species is naturally being grown in countries such as China, Nepal and Pakistan [8]. The widespread application of these plants is due to the constituents of essential oils found within them, which have rare insecticidal, antimicrobial and antiparasitic properties [9,10]. The species was studied as an allelopathic plant, and its biomass feedstock for the food industry, phytochemistry or pharmacology. Fibrous materials are also found in biomass [11,12,13]. In Lithuania, *A. dubia* began to be studied as an energy crop relatively recently. Here, some agronomic and energetic issues were evaluated.

Nitrogen, as a vital nutrient element, on plant productivity has been studied in numerous studies [14,15]. In this respect, the influence of mineral (particularly nitrogen) and organic fertilizers on *A. dubia* productivity was studied at two different locations in Lithuania. The obtained results show that the influence of nitrogen fertilizers on *A. dubia* productivity varies greatly and depends on the soil characteristics of the studied area. *A. dubia* productivity is also influenced by the year of growth and the climatic regime [16,17]. Nevertheless, the obtained results show that *A. dubia* is a high-yielding species and its productivity is comparable to that of other coarse plants, such as *Sida hermaphrodita* and *Miscanthus giganteus* [18,19]. Recently, field experiments with *A. dubia* also began in western Lithuania (LAMMC Vėžaičiai Branch). However, we did not find any data on whether *A. dubia* was grown and studied as an energy crop in other countries.

From a different perspective, to not compete with traditional food crops, alternative non-food crops or energy crops might be grown on the so-called “marginal land”, which is less suitable or unprofitable for the traditional farming land [20,21]. Despite there being different soil types in western Lithuania, natural acid soils are prevailing in this region. Due to the high natural acidity of such a type of soil, the cultivation of many traditional agricultural crops is not profitable. As such, the acidic soils must be periodically limed or fertilized by organic substances [22]. Alongside that, some land plots may be designated for energy crop cultivation. According to the above-mentioned scientific sources, *A. dubia* is a high-yielding species. No data were found on how to successfully grow *A. dubia* and what their productivity is when growing them in acidic soil. When initiating these studies, we wanted to find out how *A. dubia* tolerates an acidic soil environment and what *A. dubia* biomass productivity is like when cultivating in *Retisol*.

Since *A. dubia* was studied as an energy crop, the second task of this work was to evaluate the energy analysis of cultivation, and assess the chemical analysis, as well as the calorific value of the biomass.

## 2. Results

### 2.1. Meteorological Conditions

In 2019, there was less rain in June and August, while in July and September, the amount of precipitation was appr. twice as high (Figure 1). The amount of precipitation per vegetation was350 mm and ATS (active temperature sums higher than 10 °C) 2419 °C. In 2020, the beginning of the growing season (April–May) was quite cool. The rainfall was also low, with the highest amount falling in June. In the following months, the quantity of rainfall decreased somewhat. The total amount of precipitation was similar to the amount of the previous year, i.e., 328 mm, and the ATS 2250 °C. In 2021, the cool period in April–May and very warm period in June–July can be distinguished. It did not rain much, but since the middle of August, it rained heavily. As for the 2021 vegetation, the annual amount of precipitation was 410 mm and the ATC was 2317 °C.

The soil moisture reserves in the soil varied greatly depending on the year. However, there was not a single year in which there was no periodic moisture deficiency in the topsoil (compared to the optimum for plant growth at a 17% topsoil moisture level). Usually, the most noticeable decrease in the moisture content was during intensive growth in the first half of vegetations. Characteristically, moisture reserves in the soil were the lowest in June. The greatest lack of moisture was observed during the 2021 vegetation period.

### 2.2. Artemisia dubia Yield Parameters, DM Yield and NUE

The data of Fisher’s criterion mean squares reveal that the stem height positively depended on factors such as growing year (Factor A), cutting time (Factor B), nitrogen fertilization (Factor C) and the interaction of the year × nitrogen rate (A × C) (*p* ≤ 0.01) (Table 1). However, the number of stems was not affected by any of the investigated factors at any level of significance. All three factors, as well as year x number of cutting time (A × B) and year × nitrogen rate (A × C) interactions, had a significant and positive impact on the mass of one stem and the DM yield (all at *p* ≤ 0.01). As a parameter, the nitrogen utilization efficiency (NUE) was significantly influenced by the year (Factor A), nitrogen fertilization (Factor B) and their mutual interaction; the remaining factors had no significant impact on the parameter. It can also be noticed that the cutting time x nitrogen application rate (B × C) interaction, as well as the interaction of all three factors (A × B × C), was not significant.

As seen from the results in Figure 2, stem height as a parameter was highly variable. In 2019, i.e., the first harvestable year, *Artemisia dubia* stem height averaged 160 cm. However, the average stem height gradually decreased during the second two growing seasons down to 142 and 136 cm, respectively, in 2020 and 2021. Concerning the time of cutting, the stem height significantly increased from 103 to 176 cm. The highest stems were observed at the beginning of vegetation (first week of October). This means that *A. dubia* plants do not produce inflorescences and their stems grow throughout the whole vegetation. All differences are significant at the 95% probability level. The application of N90 was the optimal rate for stem height. Thus, the additional use of the N90 rate after the first cutting (applying N180 annual nitrogen rate) had no significant impact on the stem height.

If we look at the data of other authors, the cutting time and nitrogen increase the stem height just slightly [19].

The year the *Artemisia dubia* was planted (in 2018), it did not form any lateral stems. Contrastingly, during the second growing year (i.e., in 2019), due to the extremely rapid spreading ability into space with the help of rhizomes, *A. dubia* reached a high number of stems which was not significantly changed during the next growing season (Figure 3). Thus, the average number of stems per m^2^ varied between 85.56 and 87.78. By delaying the time of cutting, an interesting trend appears, whereby the number of stems had a decreasing tension from 90.07 to 82.59. This indicates that due to competition among adjacent growing stems, their number tends to decrease gradually. The use of nitrogen fertilizers also had a positive impact on the stem density. However, the increase was not significant at the 95% probability level.

Research held at the Vokė Branch of the LAMMC also indicates that irrespective of the nitrogen fertilization rates, the branching ability of *A. dubia* increased during the second growing season. The species produced 75–113 stems m^−2^ stems, on average. At the end of the third growing season, the number of stems reached 107–131 stems m^−2^. However, the effect of mineral nitrogen fertilization on stem density was insufficient [16].

After reaching the maximum DM yield (10.58 t ha^−1^) increase (in 2020), the yield of *A. dubia* decreased sharply during the following year, down to 7.52 t ha^−1^ (in 2021) (Figure 4). The differences in the total DM yield between experimental years are statistically reliable at a 95% probability level. Overall, in 2020, meteorological conditions (especially the moisture content in soil) were favorable for biomass formation and its accumulation. *A. dubia* does not form florets. Consequently, DM accumulation (particularly stem growth) occurs evenly throughout the whole vegetation period. Despite higher temperatures during the 2021 growing season, the growing conditions were less favorable for the crops growing and biomass accumulation because of the serious moisture lack within the topsoil layer. Biomass accumulation was mostly slowed down by a very low soil moisture in July and the first two week of August (during intense growth). The results indicate that the DM increment highly varied in the dependence on cutting time (end of June (first cutting), middle of August (second cutting) or beginning of October (third cutting)). Naturally, the lowest DM yield was fixed at the first cutting, at 6.49 t ha^−1^ on average. When *A. dubia* biomass yield was cut at the end of the growing season (i.e., beginning of October), the average DM yield increased almost twice, i.e., up to 11.82 t ha^−1^. In comparison with unfertilized plots, the N90 annual application rate caused an increase in the DM yield up to 9.02 t ha^−1^ (by 23%) and that of N180 up to 10.64 t ha^−1^ (by 45%).

The DM yield positively correlated with stem height, with +0.67, and the mass of one stem, with +0.87. There was no mutual correlation between the yield and stem density.

The high variation between seasonal *A. dubia* biomass yield was also observed in central Lithuania (soil—sandy Endocalcari-Epihypogleyic Cambisol (CMg-p-w-can)), grown on an Endocalcari-Epihypogleyic Cambisol (CMg-p-w-can) soil (pH_KCl_~6.56). Similar to our experiment, the highest *A. dubia* productivity was recorded during the second growing season, at 22.4 to 27.0 t ha^−1^. However, the DM yield sharply decreased during the third growing year, down to 14.9–16.5 t ha^−1^ [23]. Additionally, the highest biomass yield was reached in *A. dubia* swards with a nitrogen fertilization rate of N170 kg [19].

In contrast, data from an experiment held in eastern Lithuania (soil—sandy loam *Haplic Luvisols* (pH_KCl_ 5.6–5.7)) show that when using a N90P60K90 fertilization rate, the DM yield varied from 10.15 (second growing year) to 11.13 t ha^−1^ (third growing year) [16].

The NUE (nitrogen utilization efficiency) was the lowest during the first growing year, at 3.01 (Figure 5). This can be explained by the fact that there was still a sufficient reserve of nitrogen as a nutrient in the experimental upper soil layer. In this respect, the nitrogen efficiency during the first year of growth was not very significant. The influence of nitrogen fertilizers increased significantly in the following 2 years, up to 26.35 (in 2020) and 32.25 (in 2021). Since *A. dubia* stems grow throughout the whole vegetation period, the time of harvest had a very important influence on the NUE indicator (significant at the 95% probability level). In this way, at the end of the growing season (third cutting), the NUE values were the highest (significant at the 95% probability level), at 24.16 on average. The application of both nitrogen rates (N90 and N180) had a very similar impact of 18.71 and 18.35, respectively. This means that even in the case of the highest nitrogen application (N180), the increase in above-ground biomass (DM t ha^−1^) yield growth did not show a decreasing trend.

The obtained results indicate that both fertilization rates (N90 and N180) could be used for *A. dubia* fertilization.

### 2.3. Cultivation Technology Energetical Analysis

An indirect energy input consists of energy embodied in fertilizers and herbicides. Energy equivalents are used to express the input of energy related to the manufacture of fertilizers, considering the initial energy costs [24]. The indirect energy costs (or energy input) are presented in Table 2. The share of pesticides (herbicides) is low; they constitute just 864 MJ ha^−1^. The share of mineral fertilizers is much higher. Depending on the nitrogen fertilization level (N0, N90 or N180), indirect energy inputs varied from 2384 to 10,862 MJ ha^−1^, respectively. In other words, the share of energy embedded in nitrogen fertilizers was 0, 70.1 and 87.5%. Since there are a lot of energy equivalents we can find in scientific publications, the energy equivalents of the mineral N have a tendency to decreased over the years.

By calculating the direct energy expenses of *A. dubia* growth technology, we included the energy costs of agricultural tillage operations during the experiment setup, maintenance, biomass harvesting and harvesting. The following agricultural operations were used during the establishment of the experiment: experimental plot application by herbicide Roundup and the following ploughing (in autumn 2017). The next spring, the following field operations were performed: combined tillage (cultivation with arrowing (in April 2018)), seedling planting (at the beginning of June), interrow cultivation (once in the middle of July), fertilizer spreading (not at the experiment establishment year, but during the following three years), cutting biomass and transporting it to the storage place, as well as the machinery energy expenses for these operations (Table 3). The direct energy expenses in the first year of installation of the experiment are not high. They are substantial during the harvestable years. The share of human work expenses is negligible. When evaluating the technologies from a qualitative point of view, the yield loss can reach 10–15% [25,26]. The energy equivalent for 1 litre of diesel is 36 MJ ha^−1^.

To set up the experiment, the total energy expenses constituted 2816 MJ ha^−1^. On average per one harvestable year, the energy inputs were significantly higher and varied from 7015 to 16,939 MJ ha^−1^. All the differences in the energy input are due to indirect energy costs (nitrogen fertilization).

The energy input (or energetical expenses), depending on the treatment, varied from 9.83 to 19.76 GJ ha^−1^ (Table 4). Mineral fertilizers (particularly nitrogen) increased the share of the energy input from 24.25% (without nitrogen application) to 54.98% (with N180 application).

The energy output was calculated by multiplying the calorific value (GJ ha^−1^) by the average three-year DM (t ha^−1^) yield. The application of N180 caused an increase in the energy output up to 193.71 (GJ ha^−1^) (or by 43.79% compared with unfertilized plants). Nevertheless, the EUE (energy use efficiency) decreased because of nitrogen fertilization, from 13.70 (N0) down to 9.80 (N180). Usually, the EUE is higher on bigger farms. Energy efficiency is an important factor concerning the sustainable or economic use of energy [27].

### 2.4. Energy Value and Elemental Content

As seen, the *A. dubia* biomass calorific value (MJ kg^−1^) was statistically unchanged, irrespective of the cutting time (Table 5). As the result of nitrogen (N180) application to *A. dubia*, the calorific value (both lower and higher) (MJ kg^−1^) was slightly lower. Naturally, as the result of nitrogen application, the nitrogen content in biomass was somewhat higher than that of *A. dubia* biomass obtained from unfertilized plots. It seems that there is an inverse correlation between the nitrogen content and calorific value. This trend is specific for all treatments, irrespective of the cutting time. Other authors indicated that *A. dubia* has a high caloric value; however, nitrogen application has no significant impact on its calorific value [17]. The variation in the carbon (C) and hydrogen (H) content within biomass was negligible and statistically non-significant.

## 3. Materials and Methods

The field experiment with the energy crop *Artemisia dubia* Wall. (wormwood) was conducted at the Vėžaičiai Branch of Institute of Agriculture, Lithuanian Research Centre for Agriculture and Forestry in 2018–2021. The research area is locating in western Lithuania, on the eastern edge of the seaside lowland (55°430 N, 21°270 E), 70 m above sea level. The study area is naturally an acid moraine loam, *Bathygleyic Dystric Glossic Retisol* [28], with a clay content (<0.002 mm) of 15.0%. According to local meteorological data, the average annual amount of precipitation is about 915 mm.

The topsoil characteristics prior to establishing the experiment (0–30 cm soil layer) were as follows: pH_KCl_ 4.21–4.90, total N 0.06–0.08%, organic C 1.17–1.57%, humus 2.28–3.34%.

*A. dubia* seedlings were planted in 2018. A field experiment was established according to two factorial designs. The first factor was the time of harvesting. *A. dubia* stems were harvested 3 times per vegetation: the (a) end of June; (b) middle of August and (c) beginning of October. The harvestable area (netto) of all treatments was 3.3 m^2^.

The second factor was nitrogen rates. The seedlings had 3 nitrogen levels applied (0 kg ha^−1^, 90 kg ha^−1^ and 180 kg ha^−1^ (N0, N90 and N180)). Nitrogen fertilizers (as an NPK fertilizer 6–18–34, amophos (NP 12–52) and ammonium nitrate) were applied each year, just after the beginning of plant regrowth (in April). The highest nitrogen rate (N180) was split and applied two times (N90 and N90) in April and right away after the harvesting of the first yield (at the end of June). All three N treatments were randomly allocated throughout the experiment. The number of replications of the experiment was 3.

The rate of phosphorus (applied as an NPK complex 6–18–34) and partially as amophos (NP 12–52) was 60 kg ha^−1^. The rate of potassium (K_2_O) (applied as the complex 6–18–34 fertilizer) was 60 kg ha^−1^.

After three years (i.e., 2019–2021), the following parameters were evaluated in the experiment: stem height (cm), number of stems per m^2^ and the mass of one stem (g). The mass of stems and DM yield were evaluated using a weighting method. The dry matter (DM) yield (t ha^−1^) and nitrogen utilization efficiency (NUE) were evaluated for the 2019–2021 growing seasons. The nitrogen utilization efficiency (NUE) was calculated according to the following equation: ((DM yield with applied fertilizer N rate) − (DM yield in zero-N))/N rate, where N is the amount of N in kg ha^−1^ [29].

The energy assessment of *A. dubia* growing technology was performed according to the methodology elaborated at Vytautas Magnus University Agricultural Academy. The following parameters were evaluated: direct, indirect, machinery and human labour energy expenses.

A calorimeter C 2000 (IKA, Staufen, Germany) was used as an instrument to measure the calorific value (MJ kg^−1^). The analysis was carried out using special methodology [30].

When calculating the energy costs, the following fertilizer energy equivalents were included: nitrogen—47.1; phosphorus—15.8; and potassium—9.53. The energy equivalent for herbicide is 288 MJ ha^−1^ [31].

The C (carbon), N (nitrogen) and H (hydrogen) elemental compositions of pellets were determined at the Lithuanian Energy Institute using special equipment, according to standard methodology [25,26].

The total energy yield from 1 ha was calculated by multiplying the energy value of plants by the SM yield. Energy efficiency was calculated according to the formula: EE = E_out/Ein_, where: EE—energy efficiency; E_out_—energy yield (GJ ha^−1^); and E_in_—energy consumption in GJ ha^−1^ [27].

*Statistical analysis*. The data of the parameters, including stems height, number of stems, the average stem weight, dry mass (DM) yield and nitrogen utilization efficiency (NUE), were statistically processed using an analysis of variance (*ANOVA*) as a three-factorial randomized block variant to determine the significant differences between the means and the least significant difference (*LSD_05_*), and Fisher‘s criterion (*p* < 0.05) at a 95% probability level and standard deviation (±) (for chemical parameters) were used [32].

## 4. Discussion

The present *Artemisia dubia* investigation at the Vėžaičiai Branch of the LAMMC revealed that its biomass tends to increase throughout its vegetation. This means that there were significant differences between the cutting times. For this reason, when using biomass for the production of solid biofuel (e.g., pellets), the biomass should be harvested as late as possible; in our case, it would be carried out in October. As the climate changes, the following phenomenon has been observed in western Lithuania over the recent decades: when a very small amount of precipitation falls during the first half of the growing season, the moisture reserves in the upper soil layer remain very low. A lack of moisture is often associated with a high air temperature. As a result, the plants grew and developed slowly, and the already-synthesized organic materials were broken down by the plants’ more intensive respiration. Because the *A. dubia* root system is shallow, weather conditions significantly influence the growth of the above-ground biomass during the vegetation period.

Currently, investigations with *A. dubia* as an energy crop were held at three locations in Lithuania (three LAMMC branches), using similar rates of mineral nitrogen fertilizers. The highest biomass yield was obtained at the Agricultural Institute of the LAMMC, with the prevailing *Cambisol*, which is characterized as the most productive soil in Lithuania. But from the practical point of view, such soils should hardly be designed for energy crop cultivation, since fertile soils are more appropriate for food or feed crops. Western Lithuania’s *Retisols* are relatively poor; therefore, the yield of *A. dubia* productivity was noticeably lower and approximately equal to the yield of the *A. dubia* grown at the Vokė Branch of the LAMMC (the reference mentioned in the Results section). However, besides nitrogen application, there are other traditional agricultural means to improve crop productivity in acid soils like liming or the application of organic matter [22,33]. In this direction, research with *A. dubia* is continued further.

The *Artemisia* genus species have good biofuel properties. According to the results of our other long-term experiments, we can conclude that *A. dubia* productivity exceeds the yield of another related crop, e.g., *Artemisia vulgaris*, by far. We have observed the significant biomass productivity advantage of *A. dubia* compared to *A. vulgaris* in adjacent field studies [34]. Similar results have been presented by other authors [35]. Alongside that, we can conclude that *A. dubia* may also be cultivated as an alternative crop in some less productive soils, which are less suitable for the cultivation of food crops. If compared with other perennial tall coarse species (particularly *Miscanthus giganteus* L. and *Cannabis sativa* L.), the *A. dubia* biomass calorific value is high [16,35]. Its biomass may also be used for energetical purposes, since the greenhouse emission balance is almost equal to 0 [36]. Nonetheless, research on *A. dubia* as a promising energy plant species is at the initial stage, so more detailed research on its biomass energy parameters must be carried out. Moreover, the investigation of *A. dubia* biomass different energetical parameters are also at their initial stage [23,36].

The obtained results indicate that to produce the largest possible seasonal biomass yield and prepare the largest possible amount of solid fuel, *A. dubia* stems should be cut as late as possible; under Lithuanian conditions, this would be in the first half of October. In our experiment, both N90 and N180 rates demonstrated similar results (biomass increment). The amount of nitrogen fertilizer that could be applied for *A. dubia* plants also depends on the price of the fertilizer itself and the current market demand for plant biomass as the raw source of biofuels. Again, when drought prevails during growth, fertilizing with a high rate of nitrogen fertilizers does not really pay off.

Investigations of *A. dubia* are in progress and articles on various agronomic aspects of *A.dubia* cultivation and utilization for individual types of biofuels will be published in the near future.

## Figures and Tables

**Figure 1 plants-12-02441-f001:**
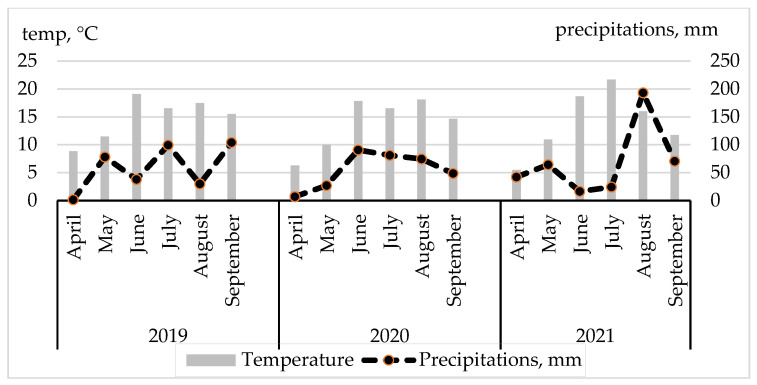
Temperature and precipitation regimes during 2019–2021 vegetations.

**Figure 2 plants-12-02441-f002:**
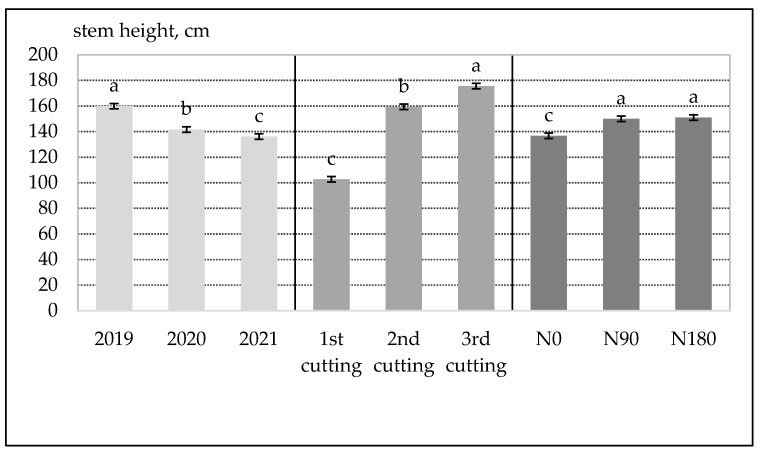
The average values of *Artemisia dubia* stem height (cm) and its dependence on the year, harvesting time (end of June, middle of August, beginning of October) and nitrogen fertilization rate (N0, N90 and N180). The bars in the columns on top correspond to the LSD_05_ values.

**Figure 3 plants-12-02441-f003:**
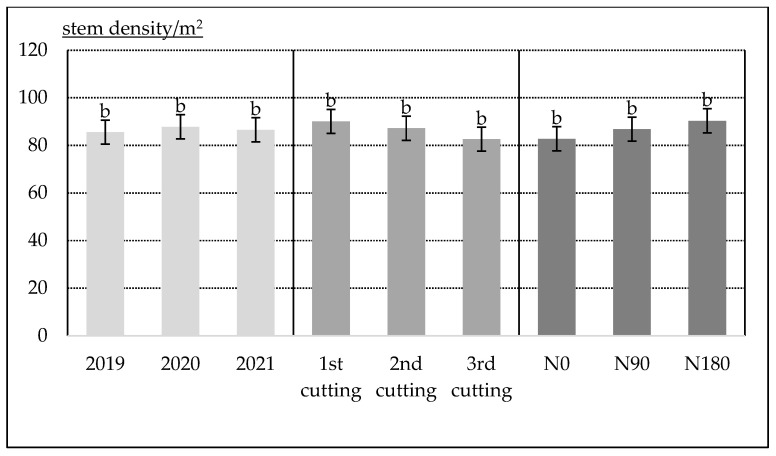
*Artemisia dubia* stem density (per m^2^) and its dependence on the year, harvesting time (end of June, middle of August, beginning of October) and nitrogen fertilization rate (N0, N90 and N180). The bars on the top of the columns correspond to the LSD_05_ values.

**Figure 4 plants-12-02441-f004:**
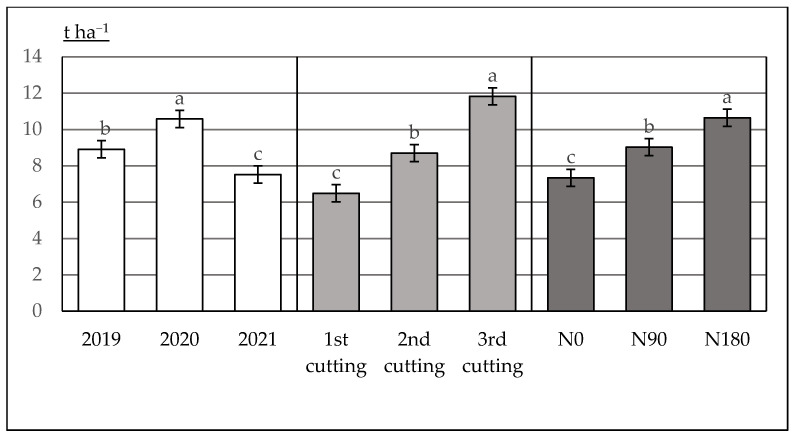
The average data of *Artemisia dubia* productivity (t ha^−1^) and its dependence on the year, end of June, middle of August, beginning of October and nitrogen fertilization rate (0, N90 and N180). The bars on top of the columns correspond to the LSD_05_ values.

**Figure 5 plants-12-02441-f005:**
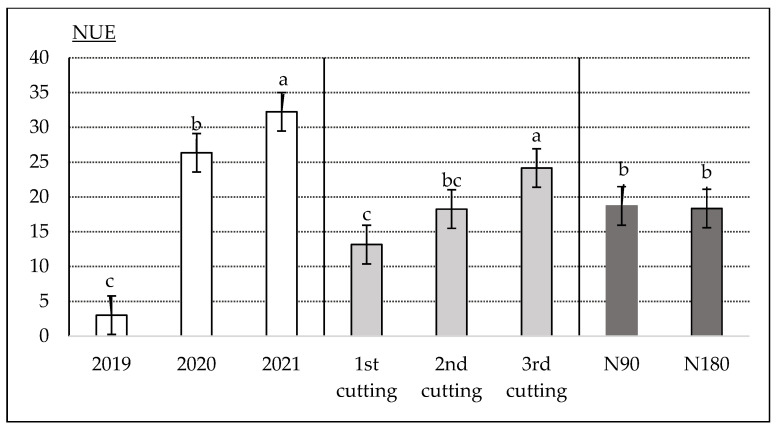
The average data of the NUE (nitrogen utilization efficiency) (kg kg^−1^) and their dependence on the cultivation year, time of harvesting and nitrogen rate. The bars on top of the columns correspond to the LSD_05_ values.

**Table 1 plants-12-02441-t001:** Fisher criterion mean squares for *Artemisia dubia* stem height, number of stems, dry matter yield (DM) and nitrogen efficiency (NUE) (for the 2019–2021 growing seasons).

Variable	Stem Height, cm	Number of Stems per m^2^	Mass of one Stem, g	DM t ha^−1^	NUE kg kg^−1^
Factor A	*	ns	*	*	*
Factor B	*	ns	*	*	ns
Factor C	*	ns	*	*	*
Interaction A × B	ns	ns	*	*	ns
Interaction A × C	*	ns	*	*	*
Interaction B × C	ns	ns	ns	ns	ns
Interaction A × B × C	ns	ns	ns	ns	ns

Note: DM—dry mater; NUE—nitrogen utilization efficiency; Factor A—year; Factor B—harvesting time; Factor C—nitrogen fertilization. * significant at *p* ≤ 0.05 level; ns—not significant.

**Table 2 plants-12-02441-t002:** The amounts of energy embodied in fertilizers and herbicides.

Herbicides, Fertilizers	The Amount Consumed for ha^−1^	Energy Equivalent MJ kg^−1^	Energy Value MJ ha^−1^
Agil	1.0 kg	288	288
Roundup	2.0 kg	288	576
N	90 kg 180 kg	47.1	4239 8478
P_2_O_5_	60 kg	15.8	948
K_2_O	60 kg	9.53	572
Total			2384—10,862

**Table 3 plants-12-02441-t003:** Energy expenses of growth technology.

Indicator	Energy Input, MJ ha^−1^		Energy Input, MJ ha ^−^ ^ 1 ^
	Experiment Setup	Harvestable Year	
Direct energy expenses	1516	4695–6096	6211–7612
Indirect energy expenses	864	1520–9998	2384–10,862
Machinery energy consumption	431	786–830	1217–1261
Human labour expenses	5.2	13.9–14.7	19.1–19.9
Total energy expenses	2816	7015–16,939	9831–19,755

**Table 4 plants-12-02441-t004:** Energy evaluation of *A. dubia* growth technology, on average.

Treatment	Energy Input GJ ha^−1^	% Share of Fertilizers	Energy Output GJ ha^−1^	EUE GJ ha^−1^
Without nitrogen	9.83	24.25	134.72	13.70
N90	14.80	44.77	164.08	11.08
N180	19.76	54.98	193.71	9.80

**Table 5 plants-12-02441-t005:** Biomass energetical parameters.

Parameter	First Cutting N0	First Cutting N180	Third Cutting N0	Third Cutting N180
Higher calorific value, MJ kg^−1^	19.12 ± 1.01	18.99 ± 0.30	19.40 ± 0.28	18.71 ± 0.25
Lower calorific value, MJ kg^−1^	17.92 ± 1.01	17.72 ± 0.30	18.14 ± 0.28	17.46 ± 0.25
C, %	48.67 ± 0.20	47.34 ± 0.12	48.73 ± 0.11	49.36 ± 0.01
N, %	1.22 ± 0.09	1.41 ± 0.13	0.68 ± 0.02	0.90 ± 0.01
H, %	5.51 ± 0.01	5.79 ± 0.06	5.76 ± 0.02	5.73 ± 0.05

## Data Availability

Not applicable.

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
