# Peer review of "The Impact of Nitrogen on the Yield Formation of Artemisia dubia Wall: Efficiency and Assessment of Energy Parameters"

_plants, 2023, doi:10.3390/plants12132441_

Round 1

Reviewer 1 Report

Dear Authors,

I find your manuscript scientifically interesting, but there are several issues that must be corrected before publication.

Introduction part is quite short, without the possibility of getting really into the topic.

Method part needs revision, sources for scientific formulas, description of the equipment and statistical software, etc. need more amplification.

Results part: There are several figures without any content, I suppose it was a formatting mistake, but e.g. fig. 2, 3, and 5 are completely blank, making the review process difficult. 

Tables need revision, e.g. table 5, there are a lot of data, you must put column lines to separate better these data.

Discussion is not scientifically deep enough, please revise it and include additional discussion part.

Kind regards

English language is fine, minor grammatical mistakes and typos are present, a final proofreading might solve this problem.

Author Response

Introduction part is quite short, without the possibility of getting really into the topic.

Indeed, Introduction was too short. We corrected and added more related text to the Chapter.

Method part needs revision, sources for scientific formulas, description of the equipment and statistical software, etc. need more amplification.

I found that indeed some methods lack scientific citations. We added two sources.

Results part: There are several figures without any content, I suppose it was a formatting mistake, but e.g. fig. 2, 3, and 5 are completely blank, making the review process difficult. 

Yes, I was also surprised that after formatting 3 figures became blank. I added again the figures and hope they will not disappear any more.

Tables need revision, e.g. table 5, there are a lot of data, you must put column lines to separate better these data.

Yes, table 5 had too much data. I did it shorter.

Discussion is not scientifically deep enough, please revise it and include additional discussion part.

That’s right. In this respect I rewrite the discussion and hope it sounds more appropriate now.

After all, thank you for the remarks. Hope the manuscript will be more engaging now.

Reviewer 2 Report

The topic is of interest and novel. However, after reading, I noticed many error which I think need to be incorprate before acceptance of this article.

1. Please revise title make it readers attractive., such as (Nitrogen fertilization is an approaching technique for the improvement of biomass energetical evaluation and productivity of Artemisia dubia

The novelty in the abstract is missing add some lines in the start.

according to the 3 (write as three) 1-10 use letter.

Please improve the conclusion of your work in the abstract section.

Introduction has some very very short paragraphs, try to avoid short paras, and also I find some old citations, try to cite recent papers and also improve the introduction section by adding more text. Below are some related citation related to fertilization, biomass and yield. try to cite it.

https://doi.org/10.3390/agronomy10030404; https://doi.org/10.1016/j.still.2018.05.007; https://doi.org/10.1016/j.agwat.2018.09.031;

Try to merge figure 2 and 3 data with table 1 (at the end).

Figure 4. I suggest to re-analysis (three way ANOVA) Year, Cutting, and Fertilization.

Also, remove the value from the top of each bar, and after three way analysis, use statistical letter on bars, to show the significant difference between Years, cutting, and Fertilization, and also show the interactive letters. By this way your figure will be meaningful. Once you have done this, then no need to show LSD value in figures.

Figure 5: No need or more to supplementary file.

Discussion is well written but short, try to add more related information.

I hope, if author can do the above comments, then the article will be improve and has chance for publication.

Author Response

Please revise title make it readers attractive., such as (Nitrogen fertilization is an approaching technique for the improvement of biomass energetical evaluation and productivity of Artemisia dubia

Yes, the title was not attractive, I think. I did the remake.

The novelty in the abstract is missing add some lines in the start.

Yes, I included the start sentence there. Also changed some sentences bellow.

according to the 3 (write as three) 1-10 use letter.

Yes, done.

Please improve the conclusion of your work in the abstract section.

Introduction has some very very short paragraphs, try to avoid short paras, and also I find some old citations, try to cite recent papers and also improve the introduction section by adding more text. Below are some related citations related to fertilization, biomass and yield. try to cite it.

I remake Introduction chapter in better way. Also included more citations from scientific articles.

Try to merge figure 2 and 3 data with table 1 (at the end).

Thank you for advice. Still, we wrote more articles in that style, so we think we can leave everything as it is now, without merging the table with the pictures.

Figure 4. I suggest to re-analysis (three way ANOVA) Year, Cutting, and Fertilization.

Indeed, I found some inconsistencies and unclear expressions there. I did corrections.

Also, remove the value from the top of each bar, and after three way analysis, use statistical letter on bars, to show the significant difference between Years, cutting, and Fertilization, and also show the interactive letters. By this way your figure will be meaningful. Once you have done this, then no need to show LSD value in figures.

It is possible. I deleted values on the top of bars. Instead of this, I put the error bars on the top.

Discussion is well written but short, try to add more related information.

Discussion chapter was rewritten and it looks different from the initial text.

Please for your revision and remarks. I hope that the quality of the manuscript was improved.

Round 2

Reviewer 1 Report

Dear Authors,

Thank you for revising the manuscript according to the reviewers' suggestions.

I still have one issue, there are no statistical marks (letters and/or asterisks) neither in the tables, nor in the figures. You should add them based on your analysis to make the differences and the conclusions clear and statistically proven.

Anyway, I have no other scientific comments on the manuscript.

Kind regards

English is fine

Author Response

Dear reviewer,

I added letters to the Figures bars. Now it looks better and more clear regarding differences between different factors. 

We did not add any letters of other asterisks to 2,3 and 4 tables, beacause these data are not experimental data. They are just calculated by choosing coefficients and energy expenses from other sources and adjusting to our results. In the similar way we aranged some articles before.

Best wishes

Gintaras Šiaudinis

Vėžaičiai Branch of the Lithuanian Research Centre for Agriculture and Forestry  
